# Development and Application of Membrane Aerated Biofilm Reactor (MABR)—A Review

Xiaolin Li [1,2,†], Dongguan Bao [3,†], Yaozhong Zhang [1,*], Weiqing Xu [1], Chi Zhang [1], Heyun Yang [1], Qiujin Ru [2], Yi-fan Wang [1], Hao Ma [1], Ershuai Zhu [1], Lianxin Dong [1], Li Li [1], Xiaoliang Li [1], Xiaopeng Qiu [1], Jiayu Tian [4] and Xing Zheng [1,*]

1   State Key Laboratory of Eco-Hydraulics in North West Arid Region, Xi'an University of Technology, Xi'an 710048, China
2   School of Hydraulic Engineering, Yang Ling Vocational & Technical College, Yangling 712100, China
3   Shanghai Hanyuan Engineering & Technology Co., Ltd., Shanghai 200000, China
4   School of Civil and Transportation Engineering, Hebei University of Technology, Tianjin 300401, China
*   Correspondence: zhangyz@xaut.edu.cn (Y.Z.); zhengxingde@yahoo.de (X.Z.); Tel.: +86-15249256622 (X.Z.)
†   The authors contribute equally to this work.

**Abstract:** As a new type of biological treatment process, membrane aerated biofilm reactors (MABRs), which have received extensive attention and research in recent years, could reduce energy consumption by 70% compared to the traditional activated sludge process. The MABR system uses bubble-free aeration membrane material as the carrier, the counter-diffusion mechanism of oxygen and pollutants enables ammonium oxidizing bacteria (AOB) and nitrate oxidizing bacteria (NOB) to adhere to the membrane surface so that simultaneous nitrification and denitrification (SND) can occur to achieve simultaneous nitrogen and carbon removal. Currently, MABR technology has been successfully applied to the treatment of municipal sewage, various industrial wastewater, pharmaceutical, high salinity, high ammonia, aquaculture wastewater, landfill leachate and black and odorous water bodies in rivers. Many laboratory experiments and pilot-scale MABR reactors have been used to study the performance of membrane materials, the mechanism of pollutant removal and the effects of different factors on the system. However, the performance of MABR is affected by factors such as dissolved oxygen (DO), pH, C/N, biofilm thickness, hydraulic retention time (HRT), temperature, etc., which limits large-scale promotion. Therefore, membrane materials, membrane modules, biofilm, application of MABR technology, influencing factors of MABR system performance, and limitations and perspectives of MABR are reviewed in this paper, and we expect to provide valuable information.

**Keywords:** membrane aerated biofilm reactor (MABR); membrane material; biological treatment process; biofilm; bubbleless aeration

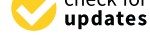



## 1. Introduction

With the increase of urbanization rate, economic development, and increase of population, global wastewater production continues to rise; global wastewater production is about 420 billion $m^3$ per year at present. On one hand, the contents of refractory pollutants in the wastewater, such as heavy metals, perfluorooctanoic acid (sulfonic acid), disinfection byproducts, endocrine disruptors, pharmaceutical and personal care products (PPCPs), antibiotics, etc., continue to increase [1–3], which are not easy to be degraded by the biological treatment process and will make microorganisms produce resistance genes and affect the biological treatment process [1,4,5]. On the other hand, the traditional activated sludge treatment process is a process of energy dissipation and pollution transfer, which is inconsistent with the concept of "carbon emission reduction and carbon neutralization" advocated worldwide [6]. In recent years, with the development of material science, a new type of breathable membrane material has been used in the wastewater treatment, called membrane aerated biofilm reactor (MABR) technology, also known as the membrane

biofilm reactor (MBfR) originated from the study of Yeh and Jenkins in 1978 [7]. Against the other membrane technologies, the breathable membrane materials used in MABR do not act as filters for wastewater treatment, but are used to transport oxygen required for wastewater treatment and carriers for biofilm growth [8]. In the MABR system, this special oxygen transfer mode and biofilm grown in the system can reduce energy consumption and floor area of treatment facilities in wastewater treatment and improve treatment efficiency [9]. At present, MABR technology has been applied to the research of small and pilot scale of domestic and various industrial wastewater, as well as the improvement of river water quality [10–12].

In the MABR system, the membrane material is assembled into a membrane component and placed in the MABR reaction tank, the aeration pump is charged into the membrane cavity through the air inlet, and the gas penetrates through the membrane material to the membrane surface [13]. Meanwhile, the biofilm attached to the membrane surface can use the penetrated gas, the gas concentration gradually decreases from the membrane surface to the outside of the biofilm, microorganisms with different functions jointly form the biofilm to achieve the purification of the target wastewater [14,15]. The common air sources are air and oxygen in MABR systems, because oxygen is necessary for nitrification process, and oxygen forms different concentration gradients in the biofilm. Therefore, the biofilm can be divided into nitrification and denitrification zones, which can realize the purpose of simultaneous nitrification and denitrification (SND) and provide conditions for the occurrence of short-cut nitrification and denitrification [16]. In addition, according to the different pollutants in the target wastewater, the gas source can also include hydrogen and methane, because hydrogen can drive microorganisms to reduce and oxidize pollutants and halogenated organics, and methane can support the synthesis and oxidation of a variety of organics [17–19].

Compared with the traditional co-diffusion biofilm system, the biofilm formed by the MABR system is called counter diffusion biofilm [20]. This aeration mode is called bubble-free aeration, the theoretical oxygen transfer efficiency is 100% [21], while the oxygen transfer efficiency of the traditional aeration method is only 15% [22,23]. Therefore, the aeration pressure required by the blower will be much smaller during operation, which provides an opportunity to reduce the energy consumption of wastewater treatment equipment during the whole life cycle to a large extent, which is of great significance while consumption in aeration contributes to beyond 50% of the energy consumption in traditional wastewater treatment plant (WWTP) [24], and such energy saving is also meaningful to the reduction of $CO_2$ emission from WWTPs. In addition, the simultaneous nitrification and denitrification existing in the MABR system can achieve smaller space and energy consumption, and can achieve the removal of more pollutants under low temperature conditions [25]. Thus, the theoretical advantages of MABR mainly focus on the following aspects: (1) the method of bubble-free aeration can greatly save the aeration energy consumption and reduce carbon emission in operation; (2) the larger specific surface area of the membrane material can attach more microorganisms, reducing the footprint of the MABR system; (3) MABR reactor can be flexibly designed, especially suitable for distributed wastewater [26,27].

Many laboratory and pilot scale MABR studies have been used to remove pollutants from various wastewater and improve river water environment [12,27,28], including domestic wastewater, oil-field wastewater, phenolic compounds in high saline wastewater, coal chemical reverse osmosis wastewater, cow manure anaerobic fermentation effluent, acetonitrile wastewater, pharmaceutical wastewater, leachate, and polluted river water [8,11,13,15,23]. In view of the wide application of MABR technology, it is necessary to comprehensively summarize the application of MABR technology in wastewater treatment. Previous reviews focus on the principle, application, and development progress of MABR technology. In this review, we expect to summarize the types of membrane materials, the forms of membrane modules, the characteristics of biological membranes, the application of this technology in different kinds of pollutants, and the influencing factors, in order to

provide a reference for researchers and engineers. Figure 1 shows the framework and key points of this overview of MABR technology.

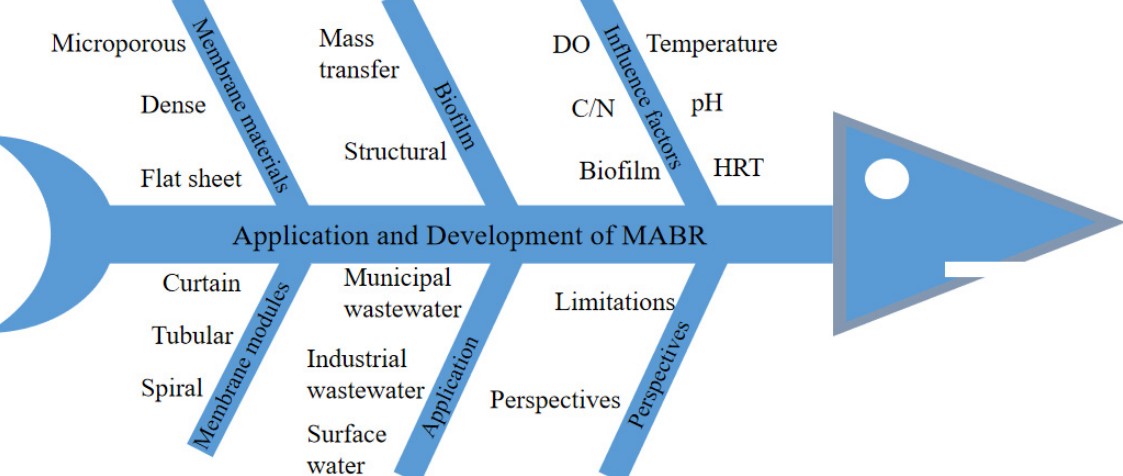

**Figure 1.** Application, development, influence factors, and perspectives of MABR.

## 2. Membrane Materials and Membrane Modules

### 2.1. Membrane Materials

In recent years, the research on permeable membrane materials has attracted the attention of companies including Suez, GE, Oxymen, and Fluence, and a variety of products have been used in practice [29,30]. As a permeable membrane material, it is required to have high oxygen permeability and good biological affinity, so as to ensure that microorganisms attach to the membrane surface to form a functional biofilm. There are mainly three kinds of common permeable membrane materials, microporous membrane, dense membrane, and flat sheet membrane [31]. The manufacturing processes of some different membrane materials are listed in Table 1, but some information is missing because there is no detailed information in the literature.

#### 2.1.1. Microporous Membrane

Hydrophobic microporous membranes are often used in MABR systems because of their cost and ease of production, with surface pore diameter of 0.01–0.2 μm [32], while hollow membrane materials are most commonly used because they can provide greater specific surface area for biological growth [33]. Most gas-permeable polymer microporous membrane are made by thermally induced phase separation (TIPS) at high temperature [34]. In the TIPS process, the semi-crystalline polymer is dissolved in the solvent at elevated temperature and causes phase separation by cooling the dope solution, and the membrane formed has highly porous and symmetrical structure [35]. However, this process needs a higher temperature, which leads to higher cost and limits its large-scale production. Moreover, the microporous membrane does not have oxygen selective permeability, and the membrane pores are easy to be blocked during operation [36].

Therefore, in order to obtain hollow membrane materials with better performance, many researchers used different methods to modify the membrane materials. Yunus and Halil modified the polyvinylidene fluoride (PVDF) microporous membrane by adding poly (vinylpyrrolidone) and propionic acid into the solvent, using the dry-jet wet spinning method, and obtained the $O_2$-based and $H_2$-based MABR membrane and used them in the MABR system, respectively [37]. The experimental results showed that the reduction rate of nitrate nitrogen and the oxidation rate of ammonia nitrogen can be improved by the $O_2$-based and $H_2$-based membrane, respectively. Wu et al. compared the two most commonly used hydrophobic microporous membranes PVDF and PP hollow fiber membranes in MABR. They found that PVDF membrane has better microbial affinity and carbon and

nitrogen removal performance, while PP membrane is prone to serious membrane pore blockage, resulting in lower oxygen transport rate than PVDF membrane [33]. Hou et al. prepared a new composite membrane by coating L-3,4-dihydroxyphenylalanine (DOPA) on the surface of PVDF membrane, it was found that the removal efficiencies of COD, $NH_4^+$-N, and TN are over 90%, 98.8%, and 84.2% in the MABR system, respectively [38]. In the process of modification, the structure and performance of the membrane can be changed by the membrane liquid composition, extrusion rate, spinneret design, air gap distance, condensation tank composition, temperature, absorption rate, and other parameters [39]. Figure 2 shows different types of gas permeable membrane materials, (A) microporous membrane [40].

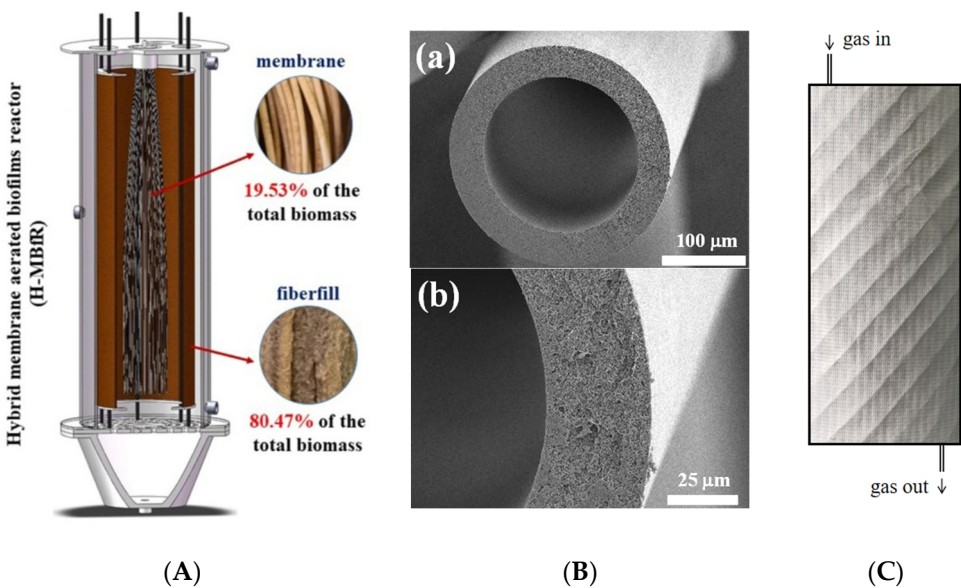

**(A)**          **(B)**          **(C)**

**Figure 2.** Different types of gas permeable membrane materials, (**A**) microporous membrane [40], (**B**) dense membrane, a, dense membrane cross section, b, membrane wall [41], (**C**) flat-sheet membrane.

### 2.1.2. Dense Membrane

Dense oxygen permeable membrane is generally made of inorganic silicone rubber, silicone resin, or ceramic materials, which has selective permeability to oxygen, high oxygen mass transfer effect and bubble point pressure, and is not easy to block compared with microporous membrane, but its manufacturing process is relatively complex and high cost [42]. Typical dense films are listed in Figure 2B. The diffusion mechanism of oxygen in dense membrane materials is dissolution diffusion, which has 100% permeability selectivity. In addition, the dense membrane made of inorganic materials has high mechanical strength and thermal stability, which can realize the separation of oxygen at low cost. Perovskite type structure is the most common structure in dense oxygen permeable membrane materials. Hendriksen et al. selected $La_{0.6}Sr_{0.4}Co_{0.2}Fe_{0.8}O_{3-\delta}$ and $SrFeCo_{0.5}O_x$ to make dense oxygen-permeable film materials and analyzed the thermodynamic stability and mechanical properties [27]. However, dense membranes also have some disadvantages, such as high pressure required to make membrane modules, complex preparation process, and high preparation cost, which limit the large-scale application of dense membrane materials.

### 2.1.3. Flat Sheet Membrane

Flat sheet membrane is generally made of composite polymer materials, such as polypropylene (PP), polyethylene (PE), polytetrafluoroethylene (PTFE), polydimethylsiloxane (PDMS), etc., which is generally made by hot pressing or coating process [33,41]. A typical flat membrane is shown in Figure 2C. Compared with microporous membrane

and dense membrane, the area of flat membrane is fixed, and the membrane itself is not self-supporting, requiring external support when used. Therefore, the application of this form of membrane in MABR is limited. The production process of flat sheet membrane is based on polymers, and then coated on its surface with a layer of permeable oxygen material, while improving the roughness of its surface. In order to improve the recovery rate of ammonia resources and treat wastewater at the same time, He et al. prepared a new flat sheet permeable membrane by paper-making and hot press process [43]. The results showed that the high-quality ammonium sulfate salt was recovered from the landfill leachate and the wastewater treatment capacity was improved. In addition, in order to improve the non-uniformity of the internal air pressure of the flat sheet membrane and the adhesion performance of the biofilm, Fluence Company designed the flat sheet membrane into a spiral membrane module, set an air distribution structure inside the membrane chamber, and added uneven diaphragms at the external interval between the membranes [44]. This design can not only make the pressure on the membrane surface uniform, but also greatly improve the adhesion of the biofilm.

To sum up, MABR membrane materials should have certain mechanical properties, high oxygen permeability, high biological affinity, high resistance to microorganisms and sewage corrosion, and long durability and price affinity.

**Table 1.** Different kinds of gas-permeable membrane materials and their manufacturing processes.

| Membrane Materials | Types of Membrane | Gas | Manufacture Process | Reference |
|---|---|---|---|---|
| L-3,4-dihydroxy-phenylalanine (DOPA)PVDF | microporous | air | coating | [38] |
| PVDF | dense | air | | [45] |
| poly(vinylpyrrolidone) or propionic acid modification PVDF | hollow fiber membrane | $O_2/H_2$ | dry-jet wet spinning | [37] |
| polyurethane or polystyrene modification polyethylene | hollow fiber | $O_2$ | coating | [41] |
| PDMS | hollow | $O_2$ | coating | [46] |
| $La_{0.6}Sr_{0.4}Co_{0.2}Fe_{0.8}O_{3-\delta}$ $SrFeCo_{0.5}O_x$ | dense | $O_2$ | high temperature calcination | [27] |
| composite materials | hollow-fiber membrane | $H_2/O_2$ | | [47] |
| nonwoven fabric substrate and porous expanded polytetraflfluoroethylene | flat sheet membrane | air | hot-press forming process | [43] |

### 2.2. Membrane Modules

According to the characteristics of membrane materials, membrane modules can be divided into curtain membrane module, tubular membrane module, flat sheet spiral membrane module, and tube spiral membrane module [18,26,30,48], as shown in Figure 3. Curtain membrane modules are generally composed of hollow membranes, which are widely used because of their high specific surface area, high oxygen enrichment capacity and good membrane dispersion (Figure 3A). Tubular membrane modules are often made of inorganic materials (Figure 3B). Spiral tubular membrane modules are generally assembled with hollow membrane materials made of polymers (Figure 3C). Due to the small specific surface area and lack of supporting capacity, the flat sheet membrane is spirally wound around a central cylinder, and uneven partitions are arranged between the diaphragm to improve its supporting capacity (Figure 3D). These forms mainly affect the gas pressure in the membrane cavity, thus affecting the gas permeation on the membrane surface.

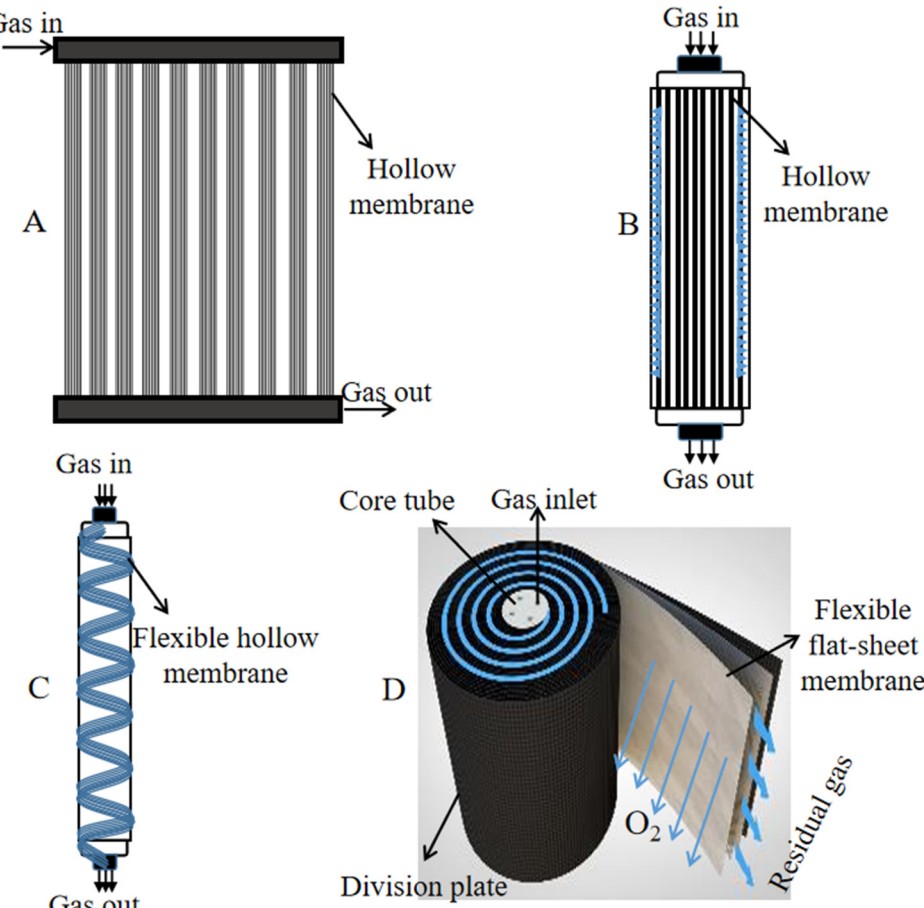

**Figure 3.** Different forms of membrane modules, (**A**) curtain membrane module, (**B**) tubular membrane module, (**C**) tube spiral membrane module, (**D**) flat sheet spiral membrane module.

In addition, membrane modules can be divided into a dead-end and cross-flow type according to different air supply modes. The dead-end membrane module consists of a hollow membrane, a gas inlet, and support for fixing the membrane module. During the operation, the membrane module is fixed in the wastewater reactor, gas enters from the gas inlet, the other side of the membrane module is sealed, and gas permeates from the surface of the tubular membrane. This operation mode has no gas loss and theoretically can achieve 100% gas utilization, so it is suitable for more expensive gases, such as $O_2$, $H_2$, and $CH_4$ [30]. However, there are two problems in this mode, when the gas supply pressure is too high, there will be gas reverse diffusion in the membrane cavity, and when the pressure is too small, the gas supply capacity will be insufficient. Thus, it is necessary to determine the suitable air supply pressure for different materials through experiments. The cross-flow membrane module is fed from one side, the gas permeates to the membrane surface through the membrane cavity, and the remaining gas is discharged from the other side. The advantage of this operation mode is that the air pressure on the surface of the membrane material can be guaranteed to be stable, and no reverse diffusion will occur, but the utilization rate of the gas is low, so it is more suitable to use air as the gas source.

Flexible plat sheet polymer membrane materials are used in spiral membrane modules. Two layers of hot melt bonded flat membrane form a sleeve with an internal airflow spacer, which is spirally wound around a core tube to allow process air to flow through their length at a pressure that is less than 10% of that required for a typical wastewater aeration diffuser. The spiral layers are separated by convex and concave partitions to ensure enough space for biofilm growth. Oxygen permeates into the biofilm on the membrane surface from the inside of the membrane cavity, and the residual gas is discharged from the end of the

spirochete. The spiral membrane module is generally cross flow type. Compared with the tubular membrane module, the spiral membrane module has a smaller specific surface area, but smaller aeration pressure is required and there is no gas back diffusion, so it is more energy-saving and more uniform membrane surface pressure [16,30].

## 3. Biofilm

### 3.1. Mass Transfer Process

As the most important part of MABR system, biofilm has higher pollutant removal efficiency than conventional carrier biofilm due to its unique substrate mass transfer process [49]. In the conventional carrier biofilm system, both oxygen and substrate permeate into the biofilm from the wastewater, and the concentration decreases continuously. This mass transfer mode is called co-diffusion system (Figure 4A). In the biofilm of the MABR system, oxygen and pollutants enter biofilm from opposite directions in the MABR system, which is called the counter-diffusion system [12] (Figure 4B). Oxygen is transferred from the membrane cavity to the inside of the biofilm on the surface of the membrane, and pollutants from the bulk liquid into the inside of the biofilm are gradually degraded [50]. The oxygen in the membrane cavity continuously enters the biofilm driven by the pressure difference. Therefore, the oxygen concentration is the highest at the biofilm membrane interface and gradually decreases and reaches a minimum at the biofilm liquid interface with the highest substrate concentration [51]. The gradient distribution of oxygen concentration makes the region close to the membrane conducive to the reproduction of nitrifying bacteria, belonging to the aerobic layer, the area away from the membrane conducive to the reproduction of denitrifying bacteria, belonging to the anaerobic layer, and the middle layer of biofilm belongs to anoxic layer [52], as shown in Figure 4. Previous studies have shown that the oxygen supply mode and the unique biofilm structure formed in MABR system can not only save energy consumption, reduce volatile gas emissions, have the function of simultaneous nitrification and denitrification, but also provide a basis for shortcut nitrification and denitrification [8,53,54].

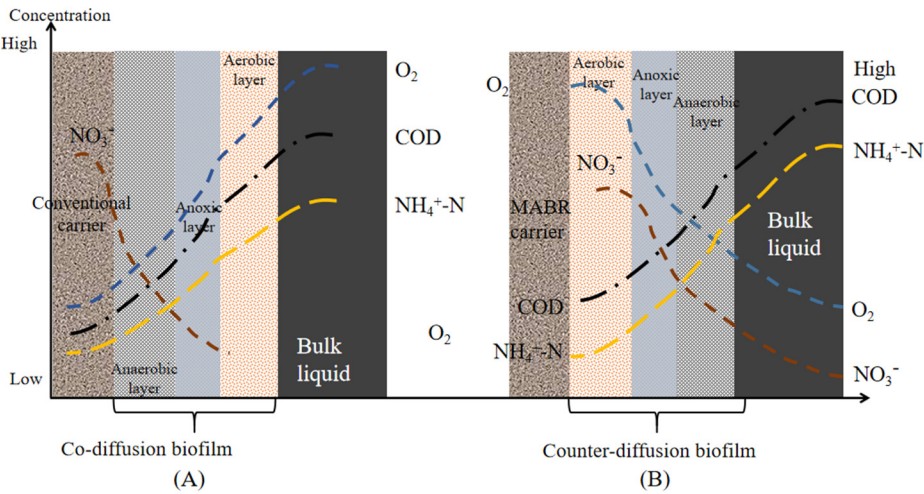

**Figure 4.** Mass transfer process of different carrier biofilm materials, (**A**) conventional carrier biofilm, (**B**) MABR carrier biofilm.

### 3.2. Biofilm Structural Characteristics

The performance of the MABR system is directly affected by the microbial community structure, and biofilm provides a habitat for microorganisms [26,49]. The biofilm can be formed naturally (surface water environment remediation) or by adding activated sludge to the MABR reactor (wastewater treatment) [55,56], which are comprised of dead and living microorganism as well as the secreted extracellular polymeric substances (EPS) [57]. Figure 5 shows the biofilm morphology observed by different methods. Due to the counter

diffusion of oxygen and substrate in the biofilm, aerobic, anoxic/anaerobic areas coexist with the biofilm, the inner side of the biofilm is an aerobic area (nitrification), and the outer side is an anoxic/anaerobic area (denitrification) [58,59]. It is well known that many factors such as growth conditions and reactor operating conditions can affect the structure and performance of biofilm [60]. For example, under the higher loading rate or substrate concentration conditions, the growth rate of biofilm is higher than the rate of erosion [61]. The biofilm surface is fluffy and mainly composed of filamentous bacteria [62]. The interior of the biofilm is denser and more erosion-resistant, and the interior of the biofilm contains more inert substances and biopolymers [63]. However, the thicker biofilm may reduce the mass transfer rate of the substrate, and anaerobic bacteria and heterotrophic bacteria multiply in large numbers, resulting in the reduction of pollutant removal efficiency and the deterioration of water quality [64]. Thinner, denser, and compact biofilm structure are formed under higher liquid velocity on the biofilm surface, the biofilm with lower viscosity fell off due to higher shear force [65,66], while the thinner biofilm may cause oxygen to penetrate the biofilm and form nitrifying biofilm, aerobic bacteria and autotrophic bacteria become dominant bacteria, reducing the denitrification performance of MABR system [67]. Therefore, in practice, feasible strategies should be formulated to keep the biofilm at the appropriate thickness.

The biofilm in MABR system provides rich ecological sites for various microorganisms, which is conducive to the removal of various refractory pollutants [68,69]. Due to the existence of oxygen concentration gradient in the biofilm, the abundance of ammonia oxidizing bacteria is higher in areas with higher oxygen and ammonia concentrations, Heterotrophic denitrifying bacteria, dechlorinating bacteria, anaerobic ammonia oxidizing bacteria, and other anaerobic bacteria are enriched in the anoxic layer and anaerobic area of the biofilm [20,52,70]. In addition, the species and abundance of microorganisms on the biofilm would be affected by the kinds of wastewater. The abundance of Proteobacteria in the outer biofilm is higher than that in the inner biofilm, when the $O_2$-MABR reactor was used to treat anaerobic fermented cow dung wastewater [23]. The dominant bacteria groups in biofilm were Proteobacteria and Bacteroidota in the MABR system for treating steel pickling rinse wastewater [71]. Various functional microorganisms on the biofilm also provide new pathway for degradation of many kinds of micropollutants. For example, Rhodococcus and Acinetobacter were found to dominate in the degradation of sulfonamide [72], while Rhodococcus was found to be related to the degradation of aromatic hydrocarbon pollutants [73]. Niastella was found to be dominant bacteria group in the treatment of tetracycline containing wastewater [74,75].

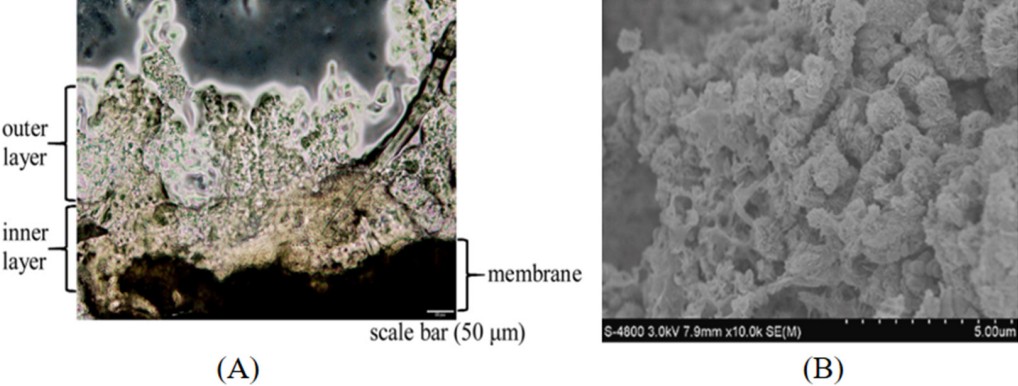

**Figure 5.** The biofilm morphology observed by different methods, (**A**) microscope [76], (**B**) scanning electron microscope [71].

## 4. Application of MABR Technology

Due to the significant advantages of MABR technology, it was first used in the treatment of domestic wastewater, and gradually expanded to all kinds of industrial wastewater and surface water environment treatment [52].

### 4.1. Application in the Municipal Wastewater Treatment

About 420 billion $m^3$ wastewater is discharged every year in the world [77]. With the continuous population increase, global wastewater production will continue to increase in the future [78]. The conventional activated sludge (AS) process is the most commonly used for domestic wastewater treatment. With the continuous increase of domestic wastewater production, the disadvantages of the conventional activated sludge process are gradually exposed, such as high energy consumption, high carbon emissions, great demand for land occupation and high sludge production, which will lead to new social problems [13]. In the past decades, with the development of material science, bubbleless aeration membrane materials have been gradually used in wastewater treatment, a number of lab-scale and pilot-scale studies have been carried out on the application of MABR technology in domestic wastewater treatment, as shown in Table 2.

Corsino et al. designed a hybrid MABR system to treat municipal sewage using Zeelung membrane material supplied by Suez; the results showed that the MABR system exhibited higher removal in terms of COD (89% vs. 85%), TN (80% vs. 65%), and TSS (86% vs. 79%) in a shorter time (1.9 days vs. 4.8 days) than conventional activated sludge [79]. In addition, the nitrification efficiency of MABR system exceeded 80% in many studies [80,81], indicating that the oxygen requirements for municipal wastewater effects can be met in MABR systems. The unique aeration mode in the MABR system promotes the formation of anammox biofilm, because this process means less energy consumption and carbon emissions in the sewage treatment process. The nitrification reaction occurs at the inner side of the MABR biofilm, while anammox biofilm is formed at the outer edge of the biofilm to promote anammox [82]. Pellicer et al. used the MABR system with continuous aeration to treat municipal wastewater. The results demonstrated that the MABR system promoted the formation of nitrite and improved the activity of anammox [83]. The above research shows that MABR system can not only meet the demand of municipal sewage treatment, but also has the potential to provide a path of low energy consumption for treatment.

**Table 2.** Summary of the application of MABR technology in domestic sewage treatment.

| Wastewater | Membrane Material | Configuration | Main Conclusion | Reference |
|---|---|---|---|---|
| Real municipal wastewater | hollow fiber membranes | Length 1 m, diameter 1.2 mm, volume 4.1 L, specific surface area 37 $m^2/m^3$ | $NH_4^+$-N, TN removal (70–90%, 60–80%) | [13] |
| Synthetic municipal wastewater | microporous polyethylene | Length 12.5 cm, volume 6 L, specific surface area 32 $m^2/m$ | Realize simultaneous nitrification and denitrification (SND) | [84] |
| Synthetic municipal wastewater | tubular PDMS membranes | Volume 0.8 L | Energy-efficient nitrogen removal with low $N_2O$ emission | [46] |
| Domestic wastewater | polyvinyl alcohol gel (PVA) | Volume 250 L | COD and TN removal 82%, 42% | [30] |
| Domestic wastewater | hollow fiber membranes | Length 1.015 m, internal diameter 200 μm, external diameter 280 μm, volume 30 L | TN, $NH_4^+$-N removal (88%, 79%) | [16] |
| Real municipal wastewater | hollow fiber membranes | volume 40 L | Nitrification (25%–40%) | [79] |
| Municipal primary effluent | Mitsubishi composite | volume 60 L | COD, TN, $NH_4^+$-N removal (74%, 80.6%, 66.7%) | [85] |
| Municipal wastewater | dense | volume 6800 L | COD, TN, $NH_4^+$-N removal (77.5%, 80.9%, 97.5%) | [81] |

### 4.2. Application in the Industrial Wastewater Treatment

The success of various small-scale and pilot-scale studies municipal wastewater treatment makes it possible for MABR technology to treat industrial wastewater. In addition, the characteristics of bubbleless aeration of MABR technology can reduce the removal of volatile organic compounds (VOCs) from the industrial wastewater, making it possible for them to be degraded by microorganisms [86]. The performance of MABR in treating different industrial wastewaters is illustrated in Table 3.

#### 4.2.1. Hospital and Pharmaceutical Wastewater

Medical and pharmaceutical wastewater is characterized by a variety of organic and chemical substances, resulting in a high load of TN, COD, and VOCs in the wastewater [10]. In addition, hospital wastewater also contains a large number of antibiotics, drug-resistant viruses, and bacteria [78]. The removal of these pollutants poses a great challenge to the traditional wastewater treatment process [87]. Tian et al. designed a pilot a MABR system consisting of hydrolysis/acidification pretreatment and activated carbon adsorption post-treatment, which was used to treat high-load mixed pharmaceutical wastewater. The effects of aeration conditions, circulating flow, and wastewater quality on system performance were studied during 260-day operation. The results showed that the oxygen utilization rate could reach 45% in the MABR combined process, and the MABR process could effectively remove more than 90% of COD and 98% of ammonia. The effluent of the combined MABR system remained stable, with COD < 200 mg/L and $NH_4^+$-N < 3 mg/L [10]. Tian et al. used MABR technology combined with ozone oxidation as a pretreatment process and the coagulation–flocculation process as a post-treatment technology to treat high-concentration pharmaceutical intermediate wastewater, the results showed that the removal of COD and TN are 98% and 91% after the pharmaceutical wastewater pre-oxidized by ozone and then passed through the MABR process [45]. This is because pre-oxidation can degrade many macromolecular substances into small molecular pollutants that are easier to be removed by MABR process.

#### 4.2.2. High Salinity and Refractory Industrial Wastewater

The pickling wastewater discharged from the steelmaking plant contains a large amount of refractory organic matter, high-load nitrogen, and salinity [88]. If the treatment is not timely and insufficient, it will cause secondary pollution to the ecological environment. Some methods such as adsorption [89], coagulation–flocculation [90], catalytic redox [91], and ion resin exchange [92] have been used for the treatment of pickling wastewater. However, the application of these methods has been severely limited due to the problems of economic cost and secondary pollution [93]. As a new biological treatment technology, MABR can be used for the treatment of pickling wastewater in iron and steel plants. Sun et al. designed a small two-stage MABR system to treat pickling wastewater containing high salinity and refractory organics. The effects of salinity and aeration pressure on the treatment effect of the system were studied, respectively. The optimal removal rate of COD, $NH_4^+$-N, and TN were 62.84%, 99.57%, and 51.65%, respectively. It is easier to achieve short nitrification and denitrification under low aeration conditions, and salinity less than 4% does not significantly affect the system performance [71]. A laboratory scale three-stage MABR system was used to treat coal chemical reverse osmosis concentrate, they studied the effects of influent salinity and different operating parameters (pH, DO, and HRT) on the treatment effect, and the results showed that under the optimal operating parameters, the removal of COD, $NH_4^+$-N, and TN reached 81.01%, 92.31%, and 70.72%; treatment efficiency and microbial communities were not significantly reduced when salinity was below 3%. Furthermore, the dominant phyla were Proteobacteria and Bacteroidetes in biofilm [94]. Tian et al. used a single-stage MABR system to treat high salinity wastewater containing phenol, p-nitrophenol, and hydroquinone, the result showed that the removal of phenolic compounds was more than 95% when the salt and total phenolic was 32 g/L and 763 mg/L, respectively. Gamma-proteobacteria, actinobacteria, and betaproteobacteria

were the dominant microorganisms in the biofilm, which contributed more to the degradation of phenolic compounds [95]. The above studies demonstrated the effectiveness of MABR technology in the treatment of high salinity and refractory organic wastewater.

### 4.2.3. Treatment Landfill Leachate

Landfill leachate contains high concentrations of pollutants such as COD, ammonia nitrogen, and inorganic salts, and its degradation has always been very difficult [96]. Physicochemical processes such as advanced oxidation processes, adsorption, hydrogen peroxide, and ozone technologies are often used for landfill leachate treatment [96,97]. However, from an economic point of view, biological treatment technology is still the most efficient technology for treating landfill leachate. Syron et al. designed a MABR reactor with a volume of 60 L to treat landfill leachate. The results suggested the system achieved 80–99% nitrification and could reduce COD by about 200–500 mg/L at the hydraulic retention time (HRT) of 5 days after one year operation, meanwhile, the oxygen transfer rate could reach 35 g $O_2$/m$^2$·day in the membrane module [8]. Therefore, the authors provided an option for MABR for low-energy-consumption treatment of landfill leachate.

### 4.2.4. Livestock Wastewater

Livestock wastewater contains a large number of pollutants and inorganic nutrients, reflecting a low C/N and low carbon–phosphorus ratio [23,98]. At present, the common treatment strategy is to use the energy in the aquaculture wastewater, such as anaerobic fermentation, to produce biogas [99]. However, the existence of anaerobic fermentation broth will still threaten the ecological environment [100]. Gong et al. designed a 1.8 L MABR reactor with pure oxygen aeration and used the gradient dilution strategy to treat the anaerobic fermentation wastewater of cow dung. The results demonstrated that the MABR system achieved higher than 88%, 87%, 93%, and 92% removal of COD, TOC, $NH_4^+$-N, and TN, respectively. In addition, they also found that the removal of $NH_4^+$-N is a complete-nitrification process at a higher dilution ratio, while the short-nitrification process is at a lower dilution ratio [23]. The MABR system can also be used for the treatment of pig manure wastewater, Terada et al. designed the MABR system which could achieve 96% and 83% removal of TOC and TN in pig manure wastewater. The biofilm thickness was 1600 μm measured by the microelectrodes; FISH analysis revealed that (AOB) was mainly distributed on the outside of the biofilm, while other microorganisms were distributed on the inside of the biofilm. The conclusions indicated that nitrification and denitrification reactions could occur simultaneously in the MABR system [101].

### 4.2.5. Petrochemical Wastewater

Petrochemical wastewater has high toxicity, high COD, and low BOD, and it is difficult to remove the soluble oil in it [102]. Numerous treatment processes have been proposed to treat petrochemical wastewater, such as advanced oxidation process, air flotation, coagulation, electro-flocculation, etc. [103,104], but these processes have some disadvantages, such as high energy consumption, high cost, and secondary pollution [20,105]. A hybrid MABR combined with an ozone-biological activated carbon system was used to treat oil-field wastewater. The authors studied the effects of influent flow rate, aeration pressure ozone oxidation time, and activated carbon retention time on the treatment effect, respectively. The results suggested that the optimal conditions were the influent flow rate of 0.06 m/s, the aeration pressure of 0.15 Mpa, the ozone oxidation time of 15 min, and the retention time of activated carbon of 30 time. The COD, TN, oil, and $NH_4^+$-N were 45, 6.8, 2.7, and 2.8 mg/L under the optimal conditions, and the effluent met the oilfield wastewater discharge standards [46]. Two-stage series-connected MABR reactors were used to treat petrochemical wastewater. Under the condition of HRT of 10 h, the removal of phenols, organic acids, TOC, $BOD_5$, and $NH_4^+$-N were about 98, >98, 80–95, 95, and 70–90%, respectively [103]. The above studies confirmed the applicability of MABR in the treatment of petrochemical wastewater.

### 4.2.6. Formaldehyde Wastewater

As a chemical raw material, formaldehyde was widely used in chemical production as a residue in industrial wastewater, which would pose a great threat to the ecological environment and human health [106]. The biological treatment process was still the most effective among all the processes for treating formaldehyde wastewater, while there were still many limitations affecting the treatment effect [107]. The biological treatment process had a low formaldehyde tolerance concentration, which would lead to volatilization and secondary pollution during aeration, high energy consumption, and high excess sludge production [108]. Mei et al. constructed a MABR system to degrade formaldehyde and analyzed the degradation kinetics and pathway of formaldehyde in the presence of co-substrate methanol (MeOH). The experimental results demonstrated that the average removal of formaldehyde achieved 97.15% when the formaldehyde was 2.99 kg/m$^3$·day. The average removal of formaldehyde, MeOH, and COD were 99.90, 97.14, and 81.50% by the MABR system when the HRT was 4 h, the aeration pressure was 0.010 Mpa and the influent formaldehyde was 116.31 mg/L. Moreover, the degradation of formaldehyde in the system follows the first-order kinetics, formaldehyde is mainly converted into formic acid and methanol through biological disproportionation reaction [26].

### 4.2.7. Acetonitrile Wastewater

Acetonitrile is a typical organic nitrile compound with carcinogenicity and mutagenicity [109]. In recent years, the research on the degradation of acetonitrile has been mainly concentrated on the aerobic process because its degradation effect is better than that of the anaerobic process. However, the high volatility of acetonitrile and the overflow of acetonitrile from wastewater during aeration caused air pollution [110,111]. Li et al. designed a 1.42 L pure oxygen aerated MABR reactor to treat acetonitrile wastewater, the effects of surface loading rate, fluid flow velocity, and HRT on system performance were studied, and the results showed that the removal of TOC and TN were 98.6% and 83.3%, when the HRT was 30 h, the surface loading rate was 11.29 g/m$^2$·d, and fluid flow velocity was 12 cm/d, the average thickness of the biofilm was 1.6 mm in the steady state, which produced more EPS enhanced microbial attachment to the membrane surface [112]. In addition, Li et al. also used a similar MABR system to study the maximum degradation ability of acetonitrile by gradually increasing the concentration of acetonitrile in the influent [113], the research result showed that the degradation ability of the system to acetonitrile reached 21.44 g/m$^2$·d, and had the ability to degrade acrylonitrile and benzonitrile [113].

**Table 3.** Performance of MABR in treating different industrial wastewaters.

| Wastewater | Membrane Material | Scale | Main Conclusion | Reference |
|---|---|---|---|---|
| Pharmaceutical wastewater | hydrophobic polypropylene dense PVDF hollow fiber | Pilot scale 1.4 L | Removal COD 90%, NH$_4^+$-N 98% Removal COD 95% and TN 92%. | [10,45] |
| Steel pickling rinse wastewater | PVDF hollow fiber | 6 L | Removal COD 62.84%, NH$_4^+$-N 99.57%, TN 51.65% | [71] |
| Phenolic wastewater | PVDF hollow fiber | 9 L | Removal phenolic compounds 95% | [95] |
| Landfill leachate | hollow-fibre polydimethyl siloxane (PDMS) membranes | 60 L | Nitrification efficiency 80–99%, Removal 75%–80% | [8] |
| Cow manure | Polytetrafluoroethylene (PTFE) | 1.8 L | Removal COD 85%, NH$_4^+$-N 90% | [23] |
| Swine liquid | Polyethylene | 0.15 L | Removal TOC 96%, TN 83% | [101] |
| Petrochemical wastewater | PDMS/silicone | 54 L | Removal TOC 80–85%, BOD$_5$ 95%, organic acids >98%, phenol 98%, NH$_4^+$-N 70–90%. | [103] |
| Oilfield Wastewater | Composite dense hollow fiber membranes | | Removal COD 82.3%, Oil 85.7%, NH$_4^+$-N 32.1%, TN 71.9%. | [45] |
| Formaldehyde wastewater | Silicone rubber membrane Tube/PDMS | | Removal FA 99.90%, MeOH 81.50% COD 97.14%. | [26] |
| Acetonitrile wastewater | Polypropylene hollow fibers | 1.42 L | Removal TOC 98.6%, TN 83.3%. | [109] |

*4.3. Application in Surface Water Treatment*

With the rapid development of the economy and the acceleration of industrialization, a great number of pollutants are discharged into the surface water, which leads to its serious deterioration [114]. The invasion of organic pollutants such as nitrogen and phosphorus leads to decreased dissolved oxygen in surface water, leading to water eutrophication and black odor [115,116]. In recent years, many technologies have been used in the restoration of surface water bodies to restore the self-purification capacity of water bodies [117], as shown in Table 4. Among the above treatment technologies, biological treatment technology is the most effective for surface water because of its environmental friendliness, convenient operation and less secondary pollution [118]. The direct reason for the deterioration of surface water is that a large amount of dissolved oxygen is consumed due to the entry of exogenous organic pollutants.

MABR technology has been used for surface water environmental remediation due to its high-efficiency oxygen supply capacity in surface water [11]. Li et al. designed a two-stage MABR system for the remediation of polluted urban river water bodies. The results showed that the removal rates of COD and $NH_4^+$-N are 87% and 95%, respectively. The two-stage MABR system has efficient remediation performance for urban rivers [11]. A coupled hydrolytic acidification pretreatment process MABR system has been applied to the remediation of micro-polluted river water. The results demonstrated that the system had the functions of simultaneous nitrification and denitrification in MABR system, could synchronously remove COD and TN in the system, and significantly improve river water quality [11]. An electrochemical MABR system was used to degrade sulfamethoxazole (SMX) and trimethoprim (TMP) in polluted surface water. Compared with the single MABR system, the application of a power plant could increase the degradation rate of SMX and TMP by 50%, because it produced a synergistic effect of reactive oxygen species and biodegradation. The results of microbial community structure analysis showed that *Flavobacterium*, which could degrade aromatic compounds, was produced in the biofilm of the EMABR, and the antibiotic resistance genes (ARGs) were inhibited in the biofilm [119]. The MABR with nylon fiber as carrier was used to treat polluted surface water. The results exhibited that this process could effectively improve the self-purification capacity of surface water, remove COD and $NH_4^+$-N, the effluent met the level V of the environmental quality standard for surface water (GB3838-2002, China), and the biomass on the carrier was more abundant [120].

MABR technology improves the surface water quality mainly through two aspects: (i) increasing the dissolved oxygen concentration in the water body through continuous aeration to the membrane module, improving the self-purification capacity of the water body and the oxidation of ammonia nitrogen [11,56]. (ii) There are nitrifying and denitrifying bacteria on the biofilm at the same time, and COD and TN are removed through simultaneous nitrification and denitrification [121].

**Table 4.** Remediation technology and application of surface water.

| Technology | Type of Surface Water | Pollutant Indexes | Main Conclusion | Reference |
|---|---|---|---|---|
| Vertical-flow constructed wetlands+ artificial aeration | Heavily polluted river | COD (65–158 mg/L), TN (5.8–12.7 mg/L) | Intermittent aeration TN and COD removal | [122] |
| Sediment dredging | River sediment pollution | Heavy metals | Reduce the content of heavy metals in rivers | [123] |
| Enhanced flocculation, polymeric ferric sulfate, polymeric aluminum chloride, $Al_2(SO_4)_3 \cdot 18H_2O$, $Fe_2(SO_4)_3$ | Dianchi lake | Cyanobacterial blooms | polymeric ferric sulfate significant algae removal effect | [124] |
| Microbial technology | Urban polluted river | Water black-odor, low dissolved oxygen concentration | DO reached 5.0 mg/L, eliminate black odor | [118] |
| Planted floating bed system | Urban river water and sediment | Nutrients and heavy metals | Higher removal of nutrients, DO and transparency are improved | [125] |

**Table 4.** *Cont.*

| Technology | Type of Surface Water | Pollutant Indexes | Main Conclusion | Reference |
|---|---|---|---|---|
| Artificial aeration and biological zeolite | Eutrophic water bodies | Total nitrogen | TN removal (78%) | [126] |
| Novel Mass Bio system and ion exchange | Micro-polluted water bodies | $NH_4^+$-N | $NH_4^+$-N removal capacity 120 t/d | [127] |
| Microalgae technology | Synthetic wastewater, black-odorous water | Emerging contaminants | Biodegradation is effective for removing $NH_4^+$-N, ibuprofen and caffeine | [128] |
| Integrated eco-engineering | Eutrophic river waters | TP, TN, COD | TP, TN, COD removal 10.5%, 11.8% and 8.2% | [129] |

## 5. Influencing Factors of MABR System Performance

### 5.1. pH

Biofilm plays a major role in the degradation of pollutants in MABR system and is significantly sensitive to pH changes in the bulk liquid. Some studies have shown that the influent pH have significant impact on the performance of MABR in nitrification and denitrification and nitrogen removal [60]. Because different kinds of bacteria on biofilm have optimal pH range for their growth. Previous studies have shown that the optimum pH range for nitrifying bacteria growth is between 6.5 and 8.5, while the optimum pH range for denitrifying bacteria growth is 7.0–8.0 [130,131]. Salehi et al. showed that the maximum loading rate of MABR system was limited by pH, the results showed when the pH was close to 7.8 in the bulk liquid, the organic nitrogen oxidation effect of the system would be limited and the pH would continue to rise due to the $NH_3$ [131]. Tian et al. used MABR combined with an ozone pretreatment coagulation combined process to treat pharmaceutical wastewater, which can achieve 95% and 92% removal rates of COD and TN, respectively, at pH = 7.2 [45]. In addition, the nitrification and denitrification reactions in the biofilm also affect the local pH and thus change the performance of MABR because two moles hydrogen ions are released for each mole of oxidation ammonia in the nitrification stage while in the denitrification stage, hydroxide ions are released through the reduction of nitrate or nitrite. the pH value depends on the concentration of ammonium, oxidation rate, denitrification rate, and alkalinity in the biofilm. In order to maintain the optimal performance of the MABR system in practice, the pH value of the bulk liquid should be kept neutral in the MABR reactor. In addition, the performance of the biofilm should be monitored according to the change of pH value in the bulk liquid and the operating conditions can be adjusted at any time according to its change to meet the requirements of wastewater treatment.

### 5.2. C/N Ration

The traditional biological treatment process needs to consume carbon sources in the denitrification stage to ensure the activity of heterotrophic bacteria, but shortcut nitrification–denitrification can reduce the consumption of carbon sources. Li et al. established a one-dimensional multi-population mathematical model using AQUASIM 2.1 software to simulate the influence of different factors on the performance of MABR [132], the results showed that the removal of TN was the highest at 78.9% when the C/N ratio was 3.75, when the C/N ratio was 2, there were AOB, NOB, and heterotrophic bacteria (HB) on the biofilm at the same time, resulting in the simultaneous removal of COD and TN; however, the removal of TN was less than 70% due to the limitation of carbon source, when the C/N was 7, HB was the dominant bacteria on the biofilm, resulting in the reduction of nitrification rate and the reduction of TN removal. The same conclusion was also confirmed by Ravi et al. They found that when the C/N was 3.4, the denitrification rate reached 95%, and the nitrate was only 1.6 mg/L in the effluent when the C/N ratio was reduced to 1.3, the nitrate was 17.12 mg/L in the effluent, and the denitrification rate was only 35% [53]. In addition, different simulations, experimental results and stoichiometric calculations

showed that the optimal C/N is lower than 4 in the MABR system [51]. The density of biofilm is also affected by the C/N, which affects the performance of MABR system [133].

### 5.3. Temperature

The performance of all biological treatment processes is affected by temperature reduction. Previous studies have shown that temperature greatly impacts the biofilm structure and microbial community in the MABR system [134]. The results of Liao et al. showed that the COD removal was 90% at 55 °C of the system, which was higher than 67% at 18–28 °C. This was because the biofilm was too thick at 18–28 °C, which limited the oxygen permeability and deteriorated the performance of MABR. The characteristics of biofilm showed that the biofilm was thinner (280 μm), denser and rich in polysaccharides under high-temperature conditions, while loose and thicker biofilm (1080 μm) was rich in proteins and hydrophobic substances under mesothermal conditions [134]. The results of Cao et al. showed that when the temperature was increased from 10 °C to 30 °C, the treatment effect was significantly improved in the MABR system and reached the optimum at 30 °C. When the temperature increased to 40 °C, the treatment efficiency decreased. The removal of nitrate and COD was the lowest when the temperature was 10 and 40 °C, because denitrifying bacteria belong to mesophilic bacteria; when the temperature increases or decreases, the metabolic activity of enzymes will be affected, resulting in a decrease in the removal of nitrate and COD [135]. Some previous studies have shown that the suitable temperature is 20–35 °C for nitrifying bacteria growth [136,137], so the system should be kept within the optimal temperature range to achieve the best performance in practice.

### 5.4. Biofilm Thickness

Biofilm plays an important role in the MABR system, and thickness is a typical indicator of the performance of MABR system. Oxygen and the substrates in the bulk liquid permeate and transfer inside the biofilm at the same time, and the process would be affected by different thicknesses. Oxygen would penetrate through the thinner biofilm and affect the denitrification process. At the same time, macromolecular organic matter would enter the inside of the biofilm and lead to the propagation of heterotrophic bacteria. The thicker biofilm would lead to the formation of a deeper anaerobic region, which is the obstacle to the transport of COD and ammonia and ultimately affect the nitrogen removal efficiency. Several different studies have shown optimal biofilm thickness in MABR systems. Terada et al. showed that the optimal thickness of biofilm for the simultaneous removal of nitrogen and COD was 1.6 mm [106]. The study by Matsumoto et al. showed that the biofilm thickness of 0.6–1.2 mm was suitable for SND [132]. Sanchez et al. studied the removal effect of MABR process on thirteen organic micro-pollutants; the results showed that the removal of pollutants was dependent on the thickness of biofilm, bacterial cell density, and microbial community composition; the removal of ammonium and COD increased gradually with increasing biofilm thickness, When the thickness of the biofilm reached 0.87 mm, the removal of ammonia reached >95%. The removal of COD reached >80% when the biofilm reached 1.02 mm [138]. In addition, the biofilm thickness is affected by many factors such as inflow velocity, flow shear force, aeration pressure, and mixing mode [52].

### 5.5. DO

Biofilm plays an important role in the MABR system, which contains both nitrifying and denitrifying bacteria for simultaneous nitrification and denitrification or shortcut nitrification and denitrification [28]. Different microorganisms have optimal oxygen concentration conditions for growth [80]. According to previous studies, the optimal DO concentration range for nitrifying bacteria is 0.3–4.0 mg/L [139], while the optimal concentration of denitrifying bacteria is 0.5–1.0 mg/L [140]. Therefore, DO concentration is different in different regions within the biofilm to enable different microorganisms to function. Li et al. found that the DO concentration was 7.79 mg/L in the inner side of the

biofilm, while close to 0.05 mg/L in the outer side of the organism when using the MABR system to degrade acetonitrile wastewater; in addition, it was anoxic in the bulk liquid in the reactor [118]. If DO is too high in the biofilm, nitrifying bacteria are the dominant bacteria species, which will have an adverse effect on denitrification. In contrast, if DO is too low in the biofilm, denitrifying bacteria become the dominant bacteria and affect the nitrification reaction. Therefore, it is necessary to maintain the proper aeration pressure for the membrane chamber to make the system play optimal in the MABR system.

*5.6. Hydraulic Retention Time (HRT)*

HRT is an important parameter in MABR process design, which is related to many factors such as reactor volume, treatment efficiency, investment, and operating cost. In a certain MABR system, lower HRT means higher treatment efficiency and lower operating cost. However, higher HRT could improve the simultaneous removal efficiency of COD and TN but increase the operating cost and reduce the treatment efficiency of the whole system. Many researchers have studied the effect of different HRTs on MABR performance [28]. According to Veleva et al. [103], a two-stage MABR reactor could achieve 85% TOC removal for the real petrochemical wastewater treatment when the HRT was 10 h. Li et al. used the two-stage MABR process of coupled hydrolysis–acidification process to purify urban river, which could achieve 87% and 95% removal rates of COD and $NH_4^+$-N, under 15 h HRT, respectively [11]. Most of the research results demonstrated that the suitable HRT for biofilm growth in MABR system ranges from 1 h to 15 days. Therefore, in practice, the appropriate HRT should be determined according to the influent load, pollutant concentration, influent flow rate, membrane material, etc.

## 6. Limitations and Perspectives of the MABR Technology
*6.1. Limitations of the MABR Technology*

Compared with conventional wastewater biological treatment technology, the advantage of MABR technology is to reduce operating costs and reduce carbon emissions. However, MABR technology is still a biological treatment process, which has all the disadvantages of the biological treatment process. The pollutant degradation process involves three-phase transformation process of oxygen (gas)–bulk liquid (liquid)–membrane material (solid) in the MABR system, which performance is affected by permeability of membrane material, thickness of biofilm, pH, temperature, HRT, etc. [53,59]. However, the control of the optimal biofilm thickness is still the most challenging. Although many scholars have carried out studies from different directions, such as using intermittent aeration, adjusting influent mode, microbial quorum sensing, and other methods to control the optimal biofilm thickness, it is still necessary to determine the control strategy of biofilm according to the actual situation in practice [141]. In addition, the research and mass production of membrane materials are still important factors restricting the promotion of MABR technology [9]. The unique process of MABR technology requires membrane material with good oxygen permeability, bio-affinity, and affordable prices. Although it has been shown that a variety of materials can be used as raw materials for aeration membrane, the high price limits large-scale production. The advantages of MABR technology are achieved by achieving SND within the system, while it could generate a greenhouse gases emission [142]. In view of the serious environmental hazards of $N_2O$ [142], special attention should be paid. Hence, it is of great significance to study the generation mechanism and pathway of $N_2O$ in MABR system. Finally, up to now, most of the research on MABR technology has been carried out under the conditions of laboratory or pilot scale, and the impact factors in actual wastewater are more complex, which limits the expanded design of MABR reactors. Therefore, researchers should consider all the problems that may affect the performance and be able to overcome them at the beginning of the expanded design.

*6.2. Prospect of MABR Technology*

At present, the research on MABR technology is in the stage of rapid development, and its characteristics of energy saving and consumption reduction have attracted the attention of many scholars and engineers. First of all, aeration membrane materials should be systematically studied in the future, focusing on the structure and manufacturing process of membrane materials in order to obtain materials with better air permeability, easier microbial adhesion, cheaper, and better durability. In addition, the most suitable membrane module form and manufacturing process of different membrane materials need to be further studied to determine the most suitable membrane module form of different membrane materials and ensure the durability of membrane modules. The optimal operating conditions of membrane materials (including gas pressure, flow shear force, mixing mode, intermittent aeration time) also need to be determined by experiments.

Biofilm plays an important role in MABR systems, which directly affects the performance of MABR reactor. Thus, the growth and formation process of biofilm should be studied, including the effects of operating conditions and environmental factors on biofilm growth, and the cultivation conditions for the formation of stable and efficient biofilm should be explored and optimized. Moreover, modern molecular biotechnology (metagenomics, proteomics and other molecular biological technologies) should be used to study the degradation mechanism and pathway of target pollutants, especially micro-pollutants, in biofilm, so as to clarify the functions of different microorganisms and the removal mechanism of pollutants. A mathematical model should also be used to establish a theoretical study of the relationship between biofilm properties and MABR wastewater treatment efficiency.

While giving full play to the maximum advantages of MABR technology, we should also pay attention to the emission of greenhouse gases, especially the emission of nitrous oxide. Isotope techniques, microelectrode measurements, and nitrogen balance calculation theories should be used to measure and quantify the nitrogen conversion process of MABR reactors under different conditions to determine the contribution of MABR reactors to greenhouse gas emissions. Likewise, the expansion of MABR reactor deserves attention; designers should consider all aspects during the scale-up design, including not only the form and layout of membrane modules, but also the way of influent and effluent, and the reduction of floor area. Finally, we should also consider the combination of MABR technology and other more technologies to meet different application scenarios.

**Author Contributions:** X.Z., Y.Z. and W.X. contributed to the design of article structure; C.Z., H.Y., H.M., and L.D. contributed to the collection and collation of the literature; Q.R., Y.-f.W., E.Z., and L.L. contributed to the analysis and summary of the literature; X.L. (Xiaolin Li) and D.B. finished writing the manuscript; X.Z., X.Q., X.L. (Xiaoliang Li), and J.T. made revisions to the manuscript. All authors have read and agreed to the published version of the manuscript.

**Funding:** This research was funded by the National Natural Science Foundation of China (No. 52170053; 51809210; 52100104) and the China Postdoctoral Science Foundation (No. 2018M633646XB), which are highly appreciated.

**Acknowledgments:** The research was conducted by the "Water Saving and Reuse Innovation Team," supported by the Educational Department of the Shaanxi Provincial Government under the Youth Innovation Team of Shaanxi Universities.

**Conflicts of Interest:** The authors declare that they have no known competing financial interests or personal relationships that could have appeared to influence the work reported in this paper.

## Abbreviations

| | | | |
|---|---|---|---|
| MABR | Membrane aerated biofilm reactor | PE | Polyethylene |
| AOB | Ammonium oxidizing bacteria | PTFE | Polytetrafluoroethylene |
| NOB | Nitrate oxidizing bacteria | PDMS | Polydimethylsiloxane |
| SND | Simultaneous nitrification and denitrification | EPS | Extracellular polymeric substances |
| DO | Dissolved oxygen | AS | activated sludge |
| HRT | Hydraulic retention time | VOCs | volatile organic compounds |
| WWTP | Wastewater treatment plant | C/N | Carbon–nitrogen ratio |
| TIPS | Thermally induced phase separation | FISH | Fluorescence in situ hybridization |
| PVDF | Polyvinylidene fluoride | MeOH | Methanol |
| DOPA | L-3,4-dihydroxyphenylalanine | HB | Heterotrophic bacteria |
| PP | Polypropylene | SMX | Sulfamethoxazole |
| COD | Chemical oxygen demand | TMP | Trimethoprim |
| TN | Total nitrogen | TP | Total phosphorus |

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
