# Peer review of "Development and Application of Membrane Aerated Biofilm Reactor (MABR)—A Review"

_water, doi:10.3390/w15030436_

Round 1
Reviewer 1 Report
The information presented in the manuscript, Development and Application of Membrane Aerated Biofilm Reactor (MABR)-A review, is related to the use of MABR to remove pollutants from various wastewater and improve the river water environment. This manuscript covers worthy topics and contains sufficient information. Most of the current papers are cited, information is presented in a logical order, the analysis is quite extensive, and conclusions are generally supported by the data. Following corrections and proofreading are needed to further improve the flow, readability, and quality of the manuscript.
1. The introduction part should be written in a broad sense and will discuss some major issues related to other membrane systems. The following studies can be cited below to strengthen the literature.
https://www.sciencedirect.com/science/article/pii/S0045653521038571
https://ami-journals.onlinelibrary.wiley.com/doi/full/10.1111/lam.13418
https://www.frontiersin.org/articles/10.3389/fmicb.2021.619323/full
https://www.sciencedirect.com/science/article/pii/S0013935122014402
https://www.sciencedirect.com/science/article/pii/S0045653522023037
Applications of tannic acid in membrane technologies: A review - ScienceDirect
2. At the end of the introduction part, there is a need to add some graphical abstract figures that depict the whole scenario of MABR, for example, the material used in MABR, Biofilm types, the potential application of MABR technology, and Influencing factors of MABR system performance.
3. In section 3, there is a need to give a detailed introduction to Biofilm, particularly focusing on the specific role of strain-dependent biofilm and the shelf function of biofilm.
4. In section 3, also discuss the potential side effects of microbial community structure in biofilm. Is the use of multiple microbial communities enhancing the potential activity of biofilm? Also, discuss factor in more detail that alters microbial community structure.
5. In section 5, it has been stated that different kinds of bacteria on biofilm have optimal pH ranges for their growth. In the conclusion section (section 6.2), what potential strategies can be used to tackle this challenge as nitrifying bacteria and denitrifying bacteria grown in their specific pH range? In addition, what stratifies have been adopted to maintain the optimum thickness of biofilm?
6. The factor that influences MABR system performance has been detailed discussed but the detail about the prospect of MABR technology is poorly discussed. The prospect of MABR technology should be discussed keeping in view the challenges faced by MABR system performance.
7. List of contents must be provided,
8. List of abbreviations must be added in revised version
9. Comparison of literature with each other is weak. Please improve.
10. In abstract mentioned the investigation of “mechanism, membrane materials, membrane modules, application scenarios and influencing factors of MABR technology are reviewed “, but there is no such classification in the text body.
11. At the end of the introduction except for two items of “membrane materials, membrane modules “the rest are different from the ones mentioned in the abstract.
Author Response
Dear reviewer 1
Please see the attachment,thank you!

Reviewer 2 Report
This papaer focused on the MABR, and provided some advances of this technology, it is good and suitable for the scopre of this Journal. Some comments for this MS.
1. It is better to describe the development of MABR with a graph.
2. In section 2, it will be better, if the authors can provide some reactor modules.
3. In Fig. 2, it will be better to add some pollutant removal processes. Such as ammonia, nitrite, nitrate and COD in different phases.
4. Section 3.2, add some Figures.
5. section 6.1, it is too general. Can authors pointed out the limitation from the influencing factors, operation difficulties, and limitation for practical application?
Author Response
Dear reviewer 2
Please see the attachment, thank you!

Reviewer 3 Report
This paper summarized the development and application of membrane aerated biofilm reactor. Influencing factors, limitations and perspectives of MABR system performance were discussed.
There are some aspects which can be improved before it is considered for publication. More literatures should be cited and a detailed discussion of pH influence are needed in page 14. Prospect needs more in it, as it's more of a summary and suggestion of the development.
Author Response
Dear reviewer 3
Please see the attachment, thank you!

Round 2
Reviewer 1 Report
The introduction should be written broadly and will discuss some major issues related to other membrane systems. The following studies can be cited ( 10.1016/j.chemosphere.2022.135810; 10.1016/j.cis.2020.102267).